# Brief communication: Plasma cortisol concentration is affected by lactation, but not intra-nasal oxytocin treatment, in beef cows

**Brooklyn K. Wagner**[1,2], **Alejandro E. Relling**[2], **Justin D. Kieffer**[3], **Anthony J. Parker**[2]*

1 Department of Population Health and Pathobiology, College of Veterinary Medicine, North Carolina State University, Raleigh, North Carolina, United States of America, 2 Department of Animal Sciences, The Ohio State University, Wooster, Ohio, United States of America, 3 Department of Animal Sciences, The Ohio State University, Columbus, Ohio, United States of America

* parker.1203@osu.edu

**Data Availability Statement:** All relevant data are within the paper and its Supporting Information files.

## Abstract

In mammals, including sheep and mice, lactation attenuates the hypothalamo-pituitary-adrenal axis and plasma cortisol concentration. Oxytocin, one neuropeptide present in the blood during lactation, may contribute to such stress attenuation. Providing oxytocin intra-nasally increases plasma oxytocin concentration in cattle and can be used in non-lactating cows to mirror plasma oxytocin concentration of lactating cows. Therefore, our hypothesis was that there would be no difference in plasma cortisol between non-lactating beef cows intra-nasally administered oxytocin and lactating beef cows intra-nasally treated with saline. Twenty *Bos taurus* cows were randomly allocated by lactational status to one of four treatments, in a 2×2 factorial arrangement: **1)** Non-lactating, saline (NL-S; n = 5); **2)** Non-lactating, oxytocin (NL-OXT; n = 5); **3)** Lactating, saline (L-S; n = 5); and **4)** Lactating, oxytocin (L-OXT; n = 5). Two hours pre-treatment, cows were catheterized, moved to their appropriate chute and baseline blood samples were collected at -60, -45, -30, and 0 minutes before treatments were administered. Directly following the 0-minute sample, cows were administered their intra-nasal treatment via a mucosal atomization device. Subsequently, blood was collected at 2, 4, 6, 8, 10, 20, 30, 40, 50, 60, 70, 80, 90, 100, 110, and 120 minutes. Non-lactating cows had greater (*P* = 0.02) plasma cortisol concentration compared with lactating cows. There was no lactation by treatment interactions for either plasma cortisol (*P* = 0.55) or oxytocin (*P* = 0.89) concentration. Although a treatment by time interaction was identified for oxytocin (*P* < 0.0001), there was no main effect of lactation on plasma oxytocin concentration (*P* = 0.34). Similar oxytocin and dissimilar cortisol concentration in lactating and non-lactating cows indicate that oxytocin alone cannot be responsible for reduced plasma cortisol in lactating ruminants. Further investigations are needed to elucidate alternative mechanisms that may be involved in the stress hypo-responsive condition of lactating mammals.

**Funding:** The author(s) received no specific funding for this work. This study was funded with discretionary funds allocated by Dr. Anthony Parker, the corresponding author, of The Ohio State University. Salaries support was provided by state and federal funds appropriated to the Ohio Agricultural Research and Development Center, The Ohio State University.

**Competing interests:** The authors have declared that no competing interests exist.

## Introduction

In mammals, including sheep and mice, lactation attenuates the hypothalamo-pituitary-adrenal (HPA) axis and plasma cortisol concentration [1, 2]. One important neuropeptide present in blood during lactation is oxytocin [3]. In a number of species, oxytocin attenuates the HPA axis activation [1, 4–6]; however, oxytocin can potentiate the HPA axis as well, resulting in greater plasma cortisol, in some species [7, 8]. Additionally, intra-nasal oxytocin treatment results in widespread dispersal of oxytocin within the brain in rodents and non-human primates [9, 10] and has attenuated increases in adrenocorticotropin hormone (another effector within the HPA axis) in non-human primates [11]. Given these results, supplementing oxytocin intra-nasally may benefit production cattle undergoing routine, yet stressful, husbandry procedures (e.g. transportation) via stress reduction and subsequent improvements to both production and welfare outcomes. However, the consequence of oxytocin treatment is highly dependent upon species [8], stressor [1, 4], physiological state [12], and route of administration [4, 13].

In ruminants, sufficient plasma oxytocin concentration is critical to promoting parturition and ensuring milk excretion. In dairy cows, complete milk removal requires plasma oxytocin concentration to be greater than basal concentration [14]. In addition, Ralph and Tilbrook report greater plasma oxytocin concentration and reduced plasma cortisol concentration in lactating ewes exposed to psycho-social stress for four hours compared with non-lactating ewes [1]. However, non-lactating beef cattle given oxytocin intra-nasally and exposed to restraint and isolation stress did not experience HPA axis attenuation [8]. Given that intra-nasal oxytocin supplementation results in a plasma concentration comparable to or greater than those reported in lactating ruminants [1, 15], the stressor is likely playing a substantial role [8]. Therefore, authors of the present study aimed to examine the effects of oxytocin administration independent of stressor and hypothesized that there would be no difference in plasma cortisol between non-lactating beef cows intra-nasally administered oxytocin and lactating beef cows intra-nasally treated with saline.

## Materials and methods

All experimental procedures were reviewed and approved by The Ohio State University Institutional Animal Care and Use Committee (#2017A00000012).

### Animal management and experimental design

Twenty *Bos taurus* cows of Angus genetic type (2–10 years of age, 510 ± 72 kg mean ± SEM bodyweight) were used in the present study. Lactating cows were 4 to 6 weeks into lactation. Cows were selected from the Ohio Agricultural Research and Development Center's beef herd at the Eastern Agricultural Research Station in Caldwell, Ohio. By age and lactational status, cows were randomly allocated to one of four treatment groups, in a 2×2 factorial arrangement: **1)** Non-lactating, saline (NL-S; 0.015 mL/kg of bodyweight 0.9% isotonic saline; n = 5); **2)** Non-lactating, oxytocin (NL-OXT; 0.60 IU/kg of bodyweight oxytocin; n = 5); **3)** Lactating, saline (L-S; n = 5); and **4)** Lactating, oxytocin (L-OXT; n = 3). Four cows were tested at a time (i.e., one from each treatment group), either in the morning or afternoon, over a three-day period. All cows were restrained in a cattle chute with a head bail. The calves (1–2 months of age) of lactating cows were kept in front of their dam to allow for visual and nose-to-nose contact between the calf and dam for the duration of the study.

### Catheterization, treatment administration and sampling

Two hours prior to treatment, cows were catheterized and then moved to their assigned chute area. Briefly, their heads were restrained to the side of the head chute to allow open access to

the jugular vein. A 5 cm × 5 cm area was clipped over the jugular vein and a baseline blood sample was collected. To begin catheter placement, a local anesthetic (3 mL, 2% Lidocaine Hydrochloride injectable, Vedco Inc., St. Joseph, Missouri, USA) was injected subcutaneously at the site of catheter placement. Betadine® surgical scrub (Purdue Pharma L.P., Stamford, CT, USA) was used to clean the area (minimum of three passes), followed by the removal of surgical scrub by alcohol (70% ethanol). A 14-gauge × 13.3cm indwelling intravenous catheter (Becton, Dickinson, and Co., Franklin Lakes, NJ, USA) was placed into the jugular vein. Catheters were secured to the skin with sutures and extension sets (i.v. extension set 36", International WIN Ltd, PA, USA) were secured to the top of the neck using Elastikon® (Thermo Fisher Scientific, Waltham, MA, USA). Cows were then moved to their appropriate chute and allowed to rest, during which time cows were able to move their head from side to side. Blood sampling was staggered based on the time of catheterization for each animal.

Baseline blood samples were taken prior to intra-nasal administration of oxytocin or saline treatments, at -60, -30, -15, and 0 minutes into a 10 mL syringe via jugular catheter. Directly following the 0-minute sample, the cow's head was restrained by placing the cow's head into the head bail, fitting a halter to her head, and tying the halter to the right side of the head bail. The assigned intra-nasal treatment (S, isotonic saline or OXT, Oxytocin, 20 IU/mL Vetone®, Bimeda-MTC Animal Health Inc., Ontario, Canada) was administered with a mucosal atomization device (Nasal™ Teleflex® Inc., Morrisville, NC) by the attending veterinarian. The dose rate of oxytocin (0.60 IU/kg of bodyweight) was chosen based on a previous report in beef cattle [8]. Doses ranged from 12 to 19 mL, dependent upon body weight. Half of each dose was administered into each nostril. Subsequently, 10 mL of blood was collected via the jugular vein catheter into a 10 mL syringe at 2, 4, 6, 8, 10, 20, 30, 40, 50, 60, 70, 80, 90, 100, 110, and 120 minutes. Collected blood was transferred into two Vacutainer™ tubes (lithium heparin and EDTA), inverted a minimum of eight times, and immediately placed on ice (4°C) until centrifugation. After sampling was complete, catheters were removed, and the animals were released back into the herd after visual assessment by the attending veterinarian.

## Sample analyses

All blood tubes were centrifuged at 3,000 × g at 4°C for 15 minutes, after which plasma was removed with a disposable pipette and stored at -20°C until required for analysis. Recovered plasma from lithium heparin tubes was used, along with a slightly modified commercially available RIA kit (MP Biomedicals, LLC., Solon, OH, USA), to measure plasma cortisol concentration. Following kit instructions for volume of sample (25 μL) yielded results below the normal limit of detection (< 10.0 ng/mL). Therefore, different volumes of plasma sample (25, 50, 100, and 150 μL) were evaluated. Using parallel displacement and recovery (101.59 ± 10%) it was determined that using 50 μL of sample yielded results between minimum and maximum concentration of 5.0 and 500.0 ng/mL, respectively. The intra-assay and inter-assay variations were 2.8 and 11.1%, respectively.

Oxytocin concentration were also determined using a commercially available double-antibody RIA kit (Oxytocin (Human, Rate, Mouse, Bovine) RIA Kit, Phoenix Pharmaceuticals, Inc., Burlingame, CA, USA) using plasma recovered from EDTA tubes. The inter-assay variation was 19.3% and the intra-assay variation ranged from 3.1 to 9.6%. The minimum and maximum concentrations of detection were 2.5 and 160.0 pg/mL, respectively. The analysis for plasma cortisol and oxytocin were completed using duplicate samples.

## Data analysis

Data were analyzed using MIXED procedure in SAS 9.4 Software (SAS Institute, 1999). Cow was treated as a random effect; treatment (S or OXT), lactational status (NL or L), time, and

their interaction were treated as fixed effects. Day and time of day (AM or PM) were included in the model as fixed effects; however, no effect of either day or time of day was detected and these variables were removed from the model. Repeated measures were assessed using the first-order autoregressive covariance structure. Selection of this covariance structure was based on lowest Bayesian information criterion between first-order autoregressive, compound symmetry, and heterogenous compound symmetry structures. Main effects of treatment, lactational status, and their interaction were assessed by ANOVA. In instances in which an interaction between these contrasts and time were identified ($P < 0.05$), an ANOVA was performed at each sampling time point. Two lactating cows from the oxytocin treatment (L-OXT) were removed from the study for uncharacteristically uncooperative behavior. Significance was determined at $P \leq 0.05$ and trends are reported at $0.05 < P \leq 0.10$. A log transformation was used on oxytocin data due to non-normality; results are presented as back transformed values.

## Results

The mean plasma cortisol concentration for all four treatments is displayed in Fig 1. Non-lactating cows had greater (Fig 2; $P = 0.02$) plasma cortisol concentration (25.3 ng/mL) compared

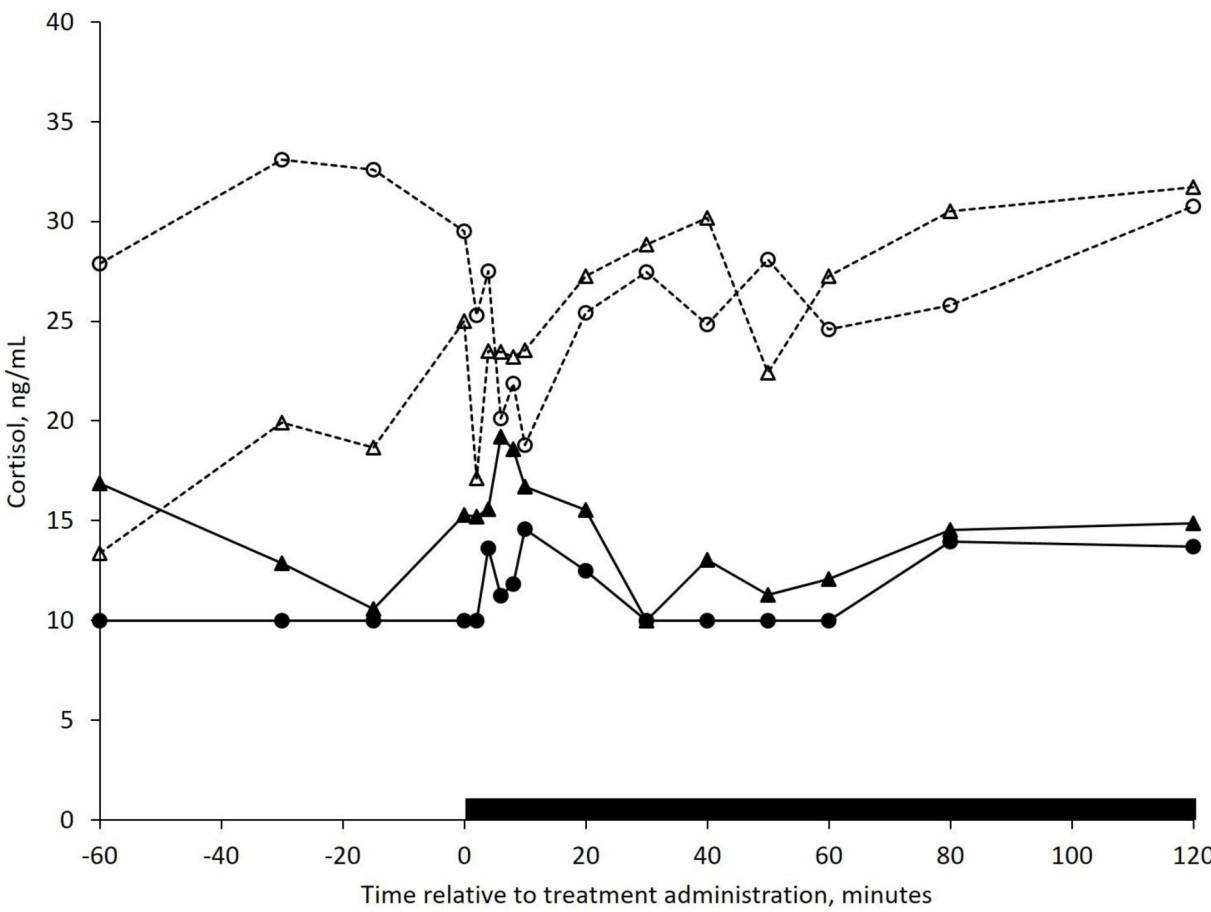

**Fig 1. Mean plasma cortisol concentrations in *Bostaurus* cows assigned to one of the following treatment. s. 1)** Non-lactating, saline (NL-S, – – Δ – –; n = 5), **2)** Non-lactating, oxytocin (NL-OXT, – – ○ – –; n = 5); **3)** Lactating, saline (L-S, ——▲——; n = 5); and **4)** Lactating, oxytocin (L-OXT, ——●——; n = 3). Intra-nasal treatments (S or OXT) were administered at 0 minutes and head restraint was applied for two hours (indicated by the black bar). Oxytocin was administered intra-nasally at a rate of 0.60 IU/kg bodyweight. The pooled SEM was 5.6. Only an effect of lactation ($P = 0.02$) was detected (see Fig 2).

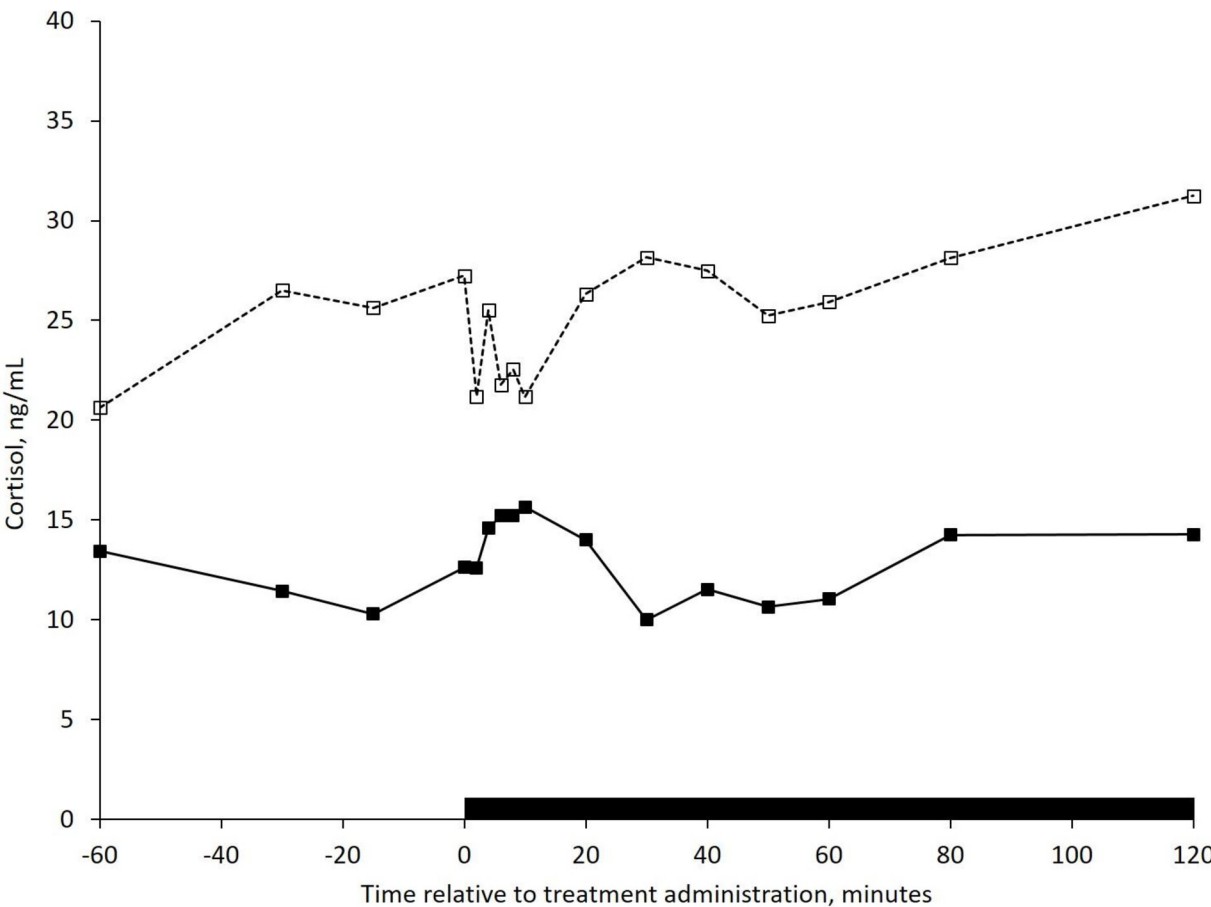

**Fig 2. Mean plasma cortisol concentration in *Bos taurus* cows.** Cows were lactating (L, —■—; n = 8) or non-lactating (NL, – – □ – –; n = 10). Head restraint start time and duration are indicated by the black bar. The pooled SEM was 4.0. An effect of lactation (*P* = 0.02) was detected, however there was no interaction between lactational status and intra-nasal treatment (*P* = 0.55).

with lactating cows (12.9 ng/mL), however no interaction between lactational status and intra-nasal treatment (*P* = 0.55) was detected. Neither intra-nasal treatment (*P* = 0.93), nor time (*P* = 0.53), affected plasma cortisol concentration.

There was no effect of lactation (*P* = 0.34) on plasma oxytocin concentration. However, an intra-nasal treatment by time interaction (Fig 3; *P* < 0.001) was detected such that oxytocin concentration increased with intra-nasal administration of oxytocin. No interactions between lactational status and intra-nasal treatment (*P* = 0.89) or time (*P* = 0.39) were detected.

## Discussion

The present study confirms that lactating beef cows with access to their calf have a reduced plasma cortisol concentration compared with non-lactating cows without a calf; and this discrepancy between cows in different physiological states is not solely a result of plasma oxytocin concentration. An attenuated response in plasma cortisol concentration has been observed previously in lactating mammals [1, 12, 16, 17], supporting lactation as a stress hypo-responsive condition. Plasma cortisol data resulting from the present study supports previous findings in ruminants [1, 17], while contradicting other reports [18]. For example, Ralph and Tilbrook [1] reported attenuated responses in plasma cortisol in sheep exposed to restraint

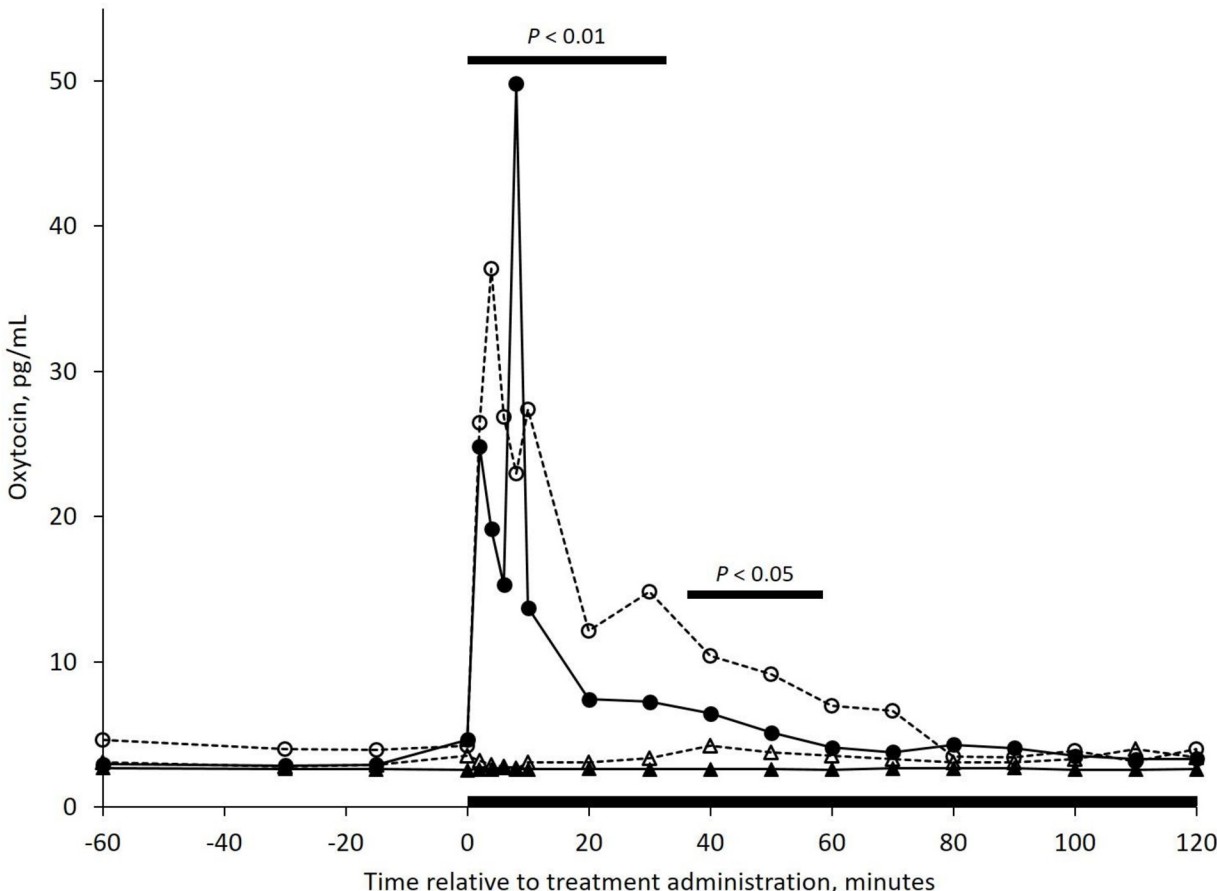

**Fig 3. Back transformed mean plasma oxytocin concentration in *Bos taurus* cows assigned to one of four treatments.** Treatments included 1) Non-lactating, saline (NL-S, – – Δ – –; n = 5), 2) Non-lactating, oxytocin (NL-OXT, – – ○ – –; n = 5); 3) Lactating, saline (L-S, —▲—; n = 5); and 4) Lactating, oxytocin (L-OXT, —●—; n = 3). Intra-nasal treatments (S or OXT) were administered at 0 minutes followed by two hours of head restraint, as indicated by the black bar. Oxytocin was administered intra-nasally at a rate of 0.60 IU/kg bodyweight. A treatment by time interaction was detected from 60 to 90 minutes ($P < 0.01$) and from 90 to 110 minutes ($P < 0.05$).

and isolation stress, while Cook [18] reported no differences in cortisol concentration between lactating and non-lactating ewes exposed to a predator stress. The present study utilized a mild head restraint in a cattle chute, with which the cows were familiar, and therefore no direct comparisons can be made between this and previous studies due to stressor differences and the stressor-dependent nature of the HPA axis response [1, 4]. Furthermore, the results of the present study support earlier published studies that state oxytocin infusion has no detectable effects on plasma cortisol concentration [8, 18].

The hypo-responsive state of the HPA axis observed in lactating mammals is likely facilitated by several hormones working in concert [2]. Oxytocin, prolactin, and the combination of oxytocin and prolactin suppress plasma cortisol concentration [2, 18, 19] and these hormones likely play a direct role in the stress-hypo-responsive state observed during lactation. However, similar plasma oxytocin concentration in both lactating and non-lactating cows in the present study indicates that lactational status alone does not support greater oxytocin concentration. Tactile stimulation of the teat occurring during suckling may be necessary to stimulate oxytocin production and release. To date and in contrast to the present study, previous investigations into the relationship between lactation and oxytocin in ruminants have allowed offspring

to suckle throughout sampling [1] which may explain discrepancies between studies. Nevertheless, similar plasma oxytocin concentration and dissimilar cortisol concentration in lactating and non-lactating cows indicate that oxytocin alone cannot be responsible for reduced plasma cortisol in lactating ruminants.

Another plausible explanation for the stress attenuation observed during lactation is that milk within the mammary gland may be acting as a sink for cortisol concentration during times of stress. Verkerk et al. [20] report increases in cortisol concentrations in the composite milk of dairy cows that coincide with a decrease in plasma cortisol concentrations. Equilibrium between plasma and alveolar milk is re-established within one hour of peak plasma cortisol concentrations [20] and an association between average cortisol concentrations in blood and milk has been established [21]. It is possible that alveolar milk may be acting as a sink for cortisol in lactating mammals. To the authors' knowledge, there are no reports of milk cortisol concentrations in beef cows.

Overall, non-lactating cows had greater plasma circulating cortisol concentration compared with lactating cows with their calf present. Additional hormones present during lactation, such as prolactin, or teat stimulation may be involved in this comparative decrease in plasma cortisol. It is also plausible that alveolar milk may be acting as a sink for cortisol in lactating mammals. Further investigations are needed to elucidate the specific mechanisms impacting the stress state of lactating mammals.

## Supporting information

**S1 Dataset.**
(XLSX)

## Author Contributions

**Conceptualization:** Brooklyn K. Wagner, Anthony J. Parker.

**Data curation:** Brooklyn K. Wagner.

**Formal analysis:** Brooklyn K. Wagner.

**Investigation:** Brooklyn K. Wagner, Anthony J. Parker.

**Methodology:** Brooklyn K. Wagner, Alejandro E. Relling, Justin D. Kieffer, Anthony J. Parker.

**Project administration:** Brooklyn K. Wagner, Justin D. Kieffer.

**Resources:** Justin D. Kieffer, Anthony J. Parker.

**Supervision:** Alejandro E. Relling, Anthony J. Parker.

**Validation:** Alejandro E. Relling.

**Visualization:** Brooklyn K. Wagner, Anthony J. Parker.

**Writing – original draft:** Brooklyn K. Wagner.

**Writing – review & editing:** Alejandro E. Relling, Anthony J. Parker.

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
