## [Decision Letter · Decision Letter 0]

8 Dec 2020

PONE-D-20-30899

Brief communication: Plasma cortisol concentration is affected by lactation, but not intra-nasal oxytocin treatment, in beef cows

PLOS ONE

Dear Dr. Parker,

Thank you for submitting your manuscript to PLOS ONE. After careful consideration, we feel that it has merit but does not fully meet PLOS ONE’s publication criteria as it currently stands. Therefore, we invite you to submit a revised version of the manuscript that addresses the points raised during the review process.

We look forward to receiving your revised manuscript.

Kind regards,

Juan J Loor

Academic Editor

PLOS ONE

Journal Requirements:

2. During your revisions, please note that a simple title correction is required: in order to avoid any confusion regarding PLOS ONE article types, we would ask that the title of your manuscript be amended to remove the phrase 'brief communication', i.e.: "Plasma cortisol concentration is affected by lactation, but not intra-nasal oxytocin treatment, in beef cows". Please ensure this is updated in the manuscript file and the online submission information.

Reviewers' comments:

Reviewer's Responses to Questions

**Comments to the Author**

1. Is the manuscript technically sound, and do the data support the conclusions?

Reviewer #1: Yes

Reviewer #2: Partly

2. Has the statistical analysis been performed appropriately and rigorously? 

Reviewer #1: Yes

Reviewer #2: Yes

3. Have the authors made all data underlying the findings in their manuscript fully available?

Reviewer #1: Yes

Reviewer #2: Yes

4. Is the manuscript presented in an intelligible fashion and written in standard English?

Reviewer #1: Yes

Reviewer #2: Yes

5. Review Comments to the Author

Reviewer #1: Authors should consider modifying Figure 1 so it contains the circulating cortisol profiles for all 4 treatments. No oxytocin treatment by lactation effects occurred but there still is merit in showing all 4 profiles. This information is, after all, what the study was designed to test.

Reviewer #2: The manuscript by Wagner et al discusses the effects of oxytocin on the cortisol response in lactating and non-lactating cows. Overall, there appears to be information missing from the Materials & Methods Section that make the progression of the experiment difficult to follow. Additionally, I have concerns about the manner in which the experiment was conducted, especially with the endpoint measurement was cortisol. Please see general and specific comments below.

Line 28: 0 minutes relative to what? It is unclear what the starting point was. It is recommended that the timing be changed where time 0 is the time of the oxytocin challenge (i.e., -60, -30, -15, 0, 2, 4, 6, etc)

Line 28: Why were cannulation and treatment/sample collection performed on the same day? Cannulation is stressful and can cause an increase in cortisol.

Line 28: Based on what is described, were cannulations/treatments/sample collection staggered throughout the day? Is there a concern of any diurnal effects on cortisol production?

Line 57: exposed to the stressor for how long?

Line 62: oxytocin administration independent of stressor

Line 77: Time of day is known to affect production and secretion of many hormones, including cortisol. Was time of day taken into consideration in the model?

Line 79: Was calf present only for lactating dams? If so, could this be a confiding factor?

Line 82: Was this 2 hours on average or was blood sampling staggered based on catherization.

Line 84: How as the baseline blood sample collected?

Line 91: Please describe extension sets.

Line 92: Please give more information on chute

Line 94: Please see previous note on changing the timeline (-60, -30, etc).

Line 95: minutes via jugular catheter

Line 95: How were the cows' heads restrained?

Line 100: 12 to 19 mL seems to be a rather large volume to be sprayed up a cow's nose. Was there any drainage after dosing?

Line 102: Blood collected via syringe and then transferred? Please describe.

Line 107: Were samples assayed in duplicate?

Line 108: centrifuged at -4 or 4 degrees C?

Line 109: until required - for what?

Line 122: Day and time of day must be added to the model - if only to prove they had no effect on the model.

Line 132: What is meant by 'fractious' behavior?

Line 138: remove 'either'

Line 161: reduced rather than lesser

Discussion: Overall the discussion is repetitive and could be reduced significantly. Considering the limited data, and combined Results and Discussion section may be more appropriate.

figures - Please add standard error bars to both graphs

Figure 2: is the spike in lactating oxytocin cows accurate? Observed in all three cows or was there an outlier? This deserves discussion.

6. PLOS authors have the option to publish the peer review history of their article (what does this mean?). If published, this will include your full peer review and any attached files.

Reviewer #1: No

Reviewer #2: No

---

## [Author Response · Author response to Decision Letter 0]

2 Feb 2021

Dear Editor,

We have amended the manuscript as suggested by the reviewers. We believe that the manuscript is improved by the comments of the reviewers and we thank them for their consideration of our manuscript. 

Sincerely

Dr Anthony Parker

---

## [Decision Letter · Decision Letter 1]

16 Mar 2021

Brief communication: Plasma cortisol concentration is affected by lactation, but not intra-nasal oxytocin treatment, in beef cows

PONE-D-20-30899R1

Dear Dr. Parker,

We’re pleased to inform you that your manuscript has been judged scientifically suitable for publication and will be formally accepted for publication once it meets all outstanding technical requirements.

Kind regards,

Juan J Loor

Academic Editor

PLOS ONE

Additional Editor Comments (optional):

Reviewers' comments:

Reviewer's Responses to Questions

**Comments to the Author**

1. If the authors have adequately addressed your comments raised in a previous round of review and you feel that this manuscript is now acceptable for publication, you may indicate that here to bypass the “Comments to the Author” section, enter your conflict of interest statement in the “Confidential to Editor” section, and submit your "Accept" recommendation.

Reviewer #1: All comments have been addressed

Reviewer #2: All comments have been addressed

2. Is the manuscript technically sound, and do the data support the conclusions?

Reviewer #1: Yes

Reviewer #2: Yes

3. Has the statistical analysis been performed appropriately and rigorously? 

Reviewer #1: Yes

Reviewer #2: Yes

4. Have the authors made all data underlying the findings in their manuscript fully available?

Reviewer #1: Yes

Reviewer #2: Yes

5. Is the manuscript presented in an intelligible fashion and written in standard English?

Reviewer #1: Yes

Reviewer #2: Yes

6. Review Comments to the Author

Reviewer #1: (No Response)

Reviewer #2: (No Response)

7. PLOS authors have the option to publish the peer review history of their article (what does this mean?). If published, this will include your full peer review and any attached files.

Reviewer #1: No

Reviewer #2: No

---

## [Editor Report · Acceptance letter]

22 Jul 2021

PONE-D-20-30899R1 

Brief communication: Plasma cortisol concentration is affected by lactation, but not intra-nasal oxytocin treatment, in beef cows. 

Dear Dr. Parker:

I'm pleased to inform you that your manuscript has been deemed suitable for publication in PLOS ONE. Congratulations! Your manuscript is now with our production department. 

Kind regards, 

on behalf of

Dr. Juan J Loor 

Academic Editor

PLOS ONE